# A SIMPLE APPROACH TO ADVERSARIAL ROBUSTNESS IN FEW-SHOT IMAGE CLASSIFICATION

## ABSTRACT

Few-shot image classification, where the goal is to generalize to tasks with limited labeled data, has seen great progress over the years. However, the classifiers are vulnerable to adversarial examples, posing a question regarding their generalization capabilities. Recent works have tried to combine meta-learning approaches with adversarial training to improve the robustness of few-shot classifiers. We show that a simple transfer-learning based approach can be used to train adversarially robust few-shot classifiers. We also present a method for novel classification task based on calibrating the centroid of the few-shot category towards the base classes. We show that standard adversarial training on base categories along with centroid-based classifier in the novel categories, outperforms or is on-par with state-of-the-art advanced methods on standard benchmarks such as Mini-ImageNet, CIFAR-FS and CUB datasets. Our method is simple and easy to scale, and with little effort can lead to robust few-shot classifiers.

## 1 INTRODUCTION

Few-shot learning presents the challenge of generalizing to unseen tasks with limited data. The problem is aimed at learning quickly from few examples of data, which is generally considered the hallmark of human intelligence. This is an important practical problem due to the scarce availability of fully annotated data in the real world. Researchers have shown that such a setting can be considered for various real world computer vision tasks such as image classification (Finn et al., 2017; Chen et al., 2019), object detection (Wang et al., 2020), image segmentation (Rakelly et al., 2018), face-recognition (Guo et al., 2020) and medical analysis (Maicas et al., 2018). As a result, it is of paramount importance that such safety-critical systems are reliable and robust to changes in input. Specifically in this work, we consider robustness to adversarial examples - carefully crafted perturbations using gradients that when added to inputs, *fool* the classifier. The most common method of improving robustness is by adversarial training (Goodfellow et al., 2015) which involves training on adversarial examples using adversary of choice. Traditional adversarially robust methods (Madry et al., 2018; Goodfellow et al., 2015; Szegedy et al., 2013) consider a data-rich setting where many examples are available per category. It has also been shown that adversarial generalization possibly requires significantly more data (Schmidt et al., 2018). This becomes challenging in a scenario where the end-user has access to limited amount of annotated data but is interested in building a robust few-shot classifier. Such a setting is more practical and it is important to develop methods which can work with minimal effort in the pre-deployment stage. We show from our experiments that a simple approach of finetuning the network on clean data from an adversarially robust base model can lead to significant improvement in robustness with minimal resources.

Previous works on improving robustness for few-shot classifiers focus mainly on meta-learning approaches. Here the base model is trained on adversarial examples of episodic data. We show that standard adversarial training on a large dataset is sufficient for learning a robust classifier. This makes our method simple and scalable, making the process of training robust classifiers straightforward and also creating directions to explore methods from robustness literature for few-shot setting.

It is important to understand the problem of few-shot learning in order to develop their robust counterparts. The goal in few-shot learning is to learn transferable knowledge for generalization to tasks with limited data. These have generally been partitioned into metric learning (Snell et al., 2017; Vinyals et al., 2016; Sung et al., 2018), optimization-based (Finn et al., 2017; Ravi & Larochelle,

2016) and hallucination based methods (Hariharan & Girshick, 2017; Yang et al., 2021; Antoniou et al., 2017; Wang et al., 2018). The most common work among optimization based methods is MAML (Finn et al., 2017) which aims at learning a network initialization using a bi-level optimization procedure, that when finetuned on limited data is able to generalize to the new task. Recent works have shown that meta-learning methods can be extended to include adversarial robustness as well. (Goldblum et al., 2019; Wang et al., 2021) perform adversarial training on top of meta-learners to improve robustness significantly. However, adversarial training on its own is expensive and combining with meta-learning makes the problem computationally intensive. (Wang et al., 2021) showed that there exists a compromise between training robust meta-learners and performance, both in terms of standard and robust accuracy, motivating the need for a simpler approach.

Another interesting line of work, focused on improving few-shot learning, shows that a simple method which involves training on large scale data and finetuning the model on the few-shot dataset can match or even outperform meta-learning methods (Chen et al., 2019; Dhillon et al., 2019). The intuition is that the model sees examples from all categories and can get a general sense of semantics rather than seeing only episodic data. Such a setting also makes it easier to train on large scale data that can lead to further improvements as shown in (Dhillon et al., 2019). Our results show considering a simple setting can be beneficial for adversarial robustness as well.

We consider the few-shot setting and show that adversarial training along with simple nearest centroid based classifier can outperform previous methods in terms of robustness. Such a setting is practically relevant, since the adversarial training on large data needs to be done just once and robustness for few-shot classes can be achieved without creating adversarial examples for the specific task. We believe it also becomes easier to incorporate new approaches to robustness, such as verifiably robust classifiers (Gowal et al., 2018; Cohen et al., 2019) and can bring together robust methods for both large and limited dataset settings.

In the following sections, we describe our method and discuss relevant related work. We present experimental findings and implementation details in the subsequent sections followed by conclusion and directions for future work.

## 2 METHOD

Here we introduce notation and provide a description of our method. Our first objective is to learn a feature extractor $f_{\theta_b}$ and linear classifier $C_{\omega_b}$ using the abundantly-labeled base dataset $X_b$. At the next stage, when a $N$-way $K$-shot few-shot task is sampled from the novel dataset $X_n$, we use only the feature extractor $f_{\theta_b}$ and learn a new linear classifier $C_{\omega_n}$ such that it can generalize to unseen examples from the novel categories. We show that a simple approach of training a robust base model and then adapting it to novel categories can outperform previous approaches. We divide our approach into two stages: (1) Robust Base training and (2) Novel training.

### 2.1 ROBUST BASE TRAINING

Given a base dataset $X_b$ with large number of annotated examples per category, we perform adversarial training using an iterative adversary such as PGD (Madry et al., 2018). Specifically, we solve the min-max objective

$$\theta^* = \min_{\theta} \mathbb{E}_{(x,y) \in X_b} \left[ \max_{||\delta||_p < \epsilon} \mathcal{L}(\theta, x + \delta, y) \right] \tag{1}$$

Here, $\mathcal{L}(\theta, x, y)$ represents the training objective, which is commonly cross-entropy and $\theta = (\theta_b, \omega_b)$ represents the combination of the feature extractor and base classifier parameters. There are different methods for optimizing the inner maximization in Equation 1. For all our experiments we use the Projected Gradient Descent (PGD) algorithm, an iterative algorithm presented in (Madry et al., 2018) with $p = \infty$ which corresponds to finding a perturbation $\delta$ around an $\epsilon$-bounded hypercube around $x$ that maximizes the objective. Once we find the perturbation, the perturbed input is added to the training set and parameters are tuned. This method is called adversarial training and is the most widely used method to improve robustness to adversarial examples. Note that adversarial training, which is a computationally expensive procedure, needs to be performed just once using base dataset.

In the next stage, we use only clean examples which makes it practical and easy to train robust few-shot classifiers for novel tasks.

**Weight averaging:** Weight averaging (WA) has been shown as a simple way to improve generalization (Izmailov et al., 2018; Garipov et al., 2018) in deep networks as it approximates ensembling in temporal fashion and can find flatter optima in loss surface. This method has been used in adversarial training (Gowal et al., 2020; Chen et al.) for improving robustness in standard classification task. Since we are interested in using base parameters at the next stage, we perform weight averaging for only the feature extractor parameters $\theta_b$ and show this can be used for few-shot setting.

Similar to (Gowal et al., 2020), we keep a separate copy of the weights and for every iteration perform exponential moving average method $\theta_b' \leftarrow \tau\theta_b' + (1-\tau)*\theta_b$ and use $\theta_b'$ during the evaluation. We set $\tau = 0.999$ in all our experiments.

## 2.2 NOVEL TRAINING

During this stage, we consider the $N$-way, $K$-shot method as the novel task and adapt our learnt feature extractor $f_{\theta_b}$ using classifier $C_{\omega_n}$. For all our experiments, we found best results when the weights of feature extractor $f_{\theta_b}$ are frozen and not optimized during novel training. Intuitively, this can be understood not wanting the parameters of the feature extractor to be biased towards the few-shot examples. And since we are interested in learning only from clean data during the novel training, there can also be an effect of *forgetting* the robustness learnt at the base stage. This was observed in (Goldblum et al., 2019) where only the final layer was trained and rest of the parameters were frozen. During the novel training stage, we use only clean examples and not adversarial examples, making the process straightforward.

**Linear classifier:** The simplest possible baseline is to learn a linear model on top of the frozen feature extractor using the few shot examples of novel categories. As shown in our experiments, this simple baseline on its own achieves reasonable performance compared to previous approaches. This baseline also suggests that a robust base classifier corresponds to a robust novel classifier and a simple approach such as ours is enough to achieve robustness for few-shot classifiers. However, as observed in previous works (Wang et al., 2021; Goldblum et al., 2019) and in our experiments, this approach alone is not sufficient to achieve improved robustness. Interestingly, we achieve much closer results to state of the art compared to previous works using this simple baseline. One challenge associated with using few-shot data is that the model can become biased towards the specific samples and may not capture the true class distribution. Hence there is a need for more calibrated classifiers.

**Background on Distribution Calibration (DC) (Yang et al., 2021):** Recent work (Yang et al., 2021) has shown that standard accuracy of few-shot classifiers can be improved by using Distribution Calibration. They present a *free-lunch* hallucination-based method where the feature distributions of the novel categories are calibrated using the base dataset, due to the similarity between the base and novel datasets. The mean and covariance of each novel category is calibrated using the statistic of base data. They use these statistics to *hallucinate* or sample many points from a Gaussian distribution, and learn a logistic regression classifier. This simple method was shown to improve standard accuracy significantly under various settings.

**Mean Calibration and Nearest Centroid (NC):**

DC method can be computationally expensive due to the calculation of covariance matrix which can be of $\mathcal{O}(N*D^2)$ complexity where $D$ is the dimensionality of the feature space and $N$ is the number of data points in the base dataset. The covariance matrix is also expensive to store in memory. Moreover, sampling from a multivariate Gaussian with non-diagonal covariance is also expensive and can be of the order of at least $\mathcal{O}(D^{2.3})$ (Bishop, 2006). These can reduce the applicability of the approach for certain architectures.

Another aspect of using the hallucinated features is that the model can become biased to the clean features and learn a non-robust final classifier. Note that since these additional data points are generated in the feature space and not the image space, it is not possible to create adversarial versions of these features and perform adversarial training using the large set of features. This poses a problem of improving the performance during novel training without sacrificing robustness.

To overcome these problems, we present a simple method where we rely only on the calibrated mean and classify query sample using a non-parametric Nearest Centroid based algorithm. We find the nearest base-category centers to each novel training sample and then average them along with the novel training sample to obtain the new mean or centroid for the novel category, similar to (Yang et al., 2021). We do not consider the covariance matrix in our method and we find this approximation works equally well in our experiments. More formally:

$$\mu_j = \frac{1}{m+1}(z_j + \sum_{i \in \mathcal{S}_j} \mu_i^b) \tag{2}$$

where $\mu_j$ is the center for the novel category $j$ and $\mathcal{S}_j$ is the set of $m$ base category centers that are closest to $z_j$, $\mu_i^b$ is the mean of base category $i$ in the feature space. In the case of $k$-shot setting, we calculate a centroid for each sample and average them to get one centroid for each category.

At inference time, we simply find the nearest center to the query point and assign its label:

$$\hat{y} = \{y_j | \arg\max_j \tilde{\mu}_j^T \tilde{z}\} \tag{3}$$

where $\tilde{a} = \frac{a}{||a||_2}$ is the $\ell_2$ normalized version of the vector $a$ and $\hat{y}$ is the predicted category. Note that $\ell_2$ normalization is performed for both query point and centroids. Since we consider the normalized version of the vectors, euclidean distance reduces to the form in Equation 3, similar to recent works (Grill et al., 2020). Our inference can be considered similar to the Linear Classifier method exceptthat our centres ( $\mu_j$ ) are estimated with simple averaging rather than learnt using SGD. Note that similar to (Yang et al., 2021), as a preprocessing step, we transform the embeddings by taking the square root of each dimension so that their distribution gets closer to Gaussian form. It can be seen as "Tukey's Ladders of Power Transformation" (Tukey et al., 1977) with $\lambda = 0.5$.

At the inference time, the Nearest Centroid (NC) method requires less memory and computation compared to a Nearest Neighbor classifier since we only have to store and compare with one prototype per class rather than the entire training set.

## 3 RELATED WORK

**Few shot Image Classification:** Few-shot learning is a challenging problem in computer vision where the goal is rapid generalization to unseen tasks. Metric learning approaches such as (Snell et al., 2017; Vinyals et al., 2016; Sung et al., 2018) were some of the earliest approaches towards tacking this problem. (Snell et al., 2017) learn a metric space where prototypical representation of each category is utilized for classifying novel data. The prototypes are calculated to be the euclidean average over the support examples. (Ravichandran et al., 2019) showed that rather than taking the average, learning the prototypes along with the model can lead to better performance. More recently, a family of algorithms based on learning to learn (Andrychowicz et al., 2016) or meta-learning have gained considerable attention. (Ravi & Larochelle, 2016) develop an LSTM based meta-learner which learns the optimization algorithm required to train another neural network. (Finn et al., 2017; Nichol et al., 2018) create a model agnostic algorithm which aims to learn a good initialization that can be finetuned easily within a few gradient steps to adapt to new tasks. Hallucination based methods such as (Hariharan & Girshick, 2017; Wang et al., 2018; Antoniou et al., 2017) also present promising directions towards improved generalization. (Gidaris & Komodakis, 2018; Qi et al., 2018) try to directly predict the weights of the classifier for novel categories. (Yang et al., 2021) calibrate the distributions of few shot examples using the statistics of categories with larger number of examples. . However a recent line of work have questioned the need for meta-learning or metric learning approaches and showed that simple baselines which are non-episodic in nature can provide competitive performance for few-shot image classification task (Chen et al., 2019; Dhillon et al., 2019). Such line of work provide for non-sophisticated baselines and pose a question to the community to rethink the approach towards few shot learning.

**Adversarial examples:** Adversarial examples are carefully crafted perturbations designed to fool the model (Szegedy et al., 2013; Goodfellow et al., 2015; Kurakin et al., 2016). (Goodfellow et al., 2015) showed that adversarial examples can be created rather easily using the sign of single gradient step, which they called as Fast Gradient Sign Method (FGSM). The existence of such examples raises a

question regarding the generalization capabilities of Deep Neural networks while also posing a threat to practical deployment. Many defenses have been proposed to overcome this problem (Papernot et al., 2016; Xie et al., 2017; Feinman et al., 2017; Li & Li, 2016), but they have been bypassed with slight modifications to the adversary (Athalye et al., 2018; Carlini & Wagner; 2017). One of the most common approaches called adversarial training involves incorporating the adversarial examples into the training set. (Madry et al., 2018) showed that the first order adversary, based on the Projected Gradient Descent (PGD) algorithm can be used to train robust neural networks. Provable methods have developed which can provide certification on the susceptibility of an input towards adversaries (Wong & Kolter, 2018; Wong et al., 2018; Raghunathan et al., 2018; Gowal et al., 2018). A class of algorithms based on randomized smoothing (Cohen et al., 2019; Yang et al., 2020) have shown promising results in training large scale neural networks. (Zhang et al., 2019a) provide a theoretical analysis on the robustness vs accuracy trade-off, which had been studied empirically (Tsipras et al., 2018), and show that their algorithm named TRADES improves robustness compared to previous approaches. (Shafahi et al., 2019) use the gradients from the backpropagation algorithm to improve robustness with minimal cost.

**Adversarial Robustness for Few-shot classifiers:** Recent works have tried to address the problem of adversarial examples in the context of few-shot learning. (Yin et al., 2018) was one of the first works to discuss this problem. They used the FGSM adversary to create adversarial examples and optimized a meta-learner to be robust to adversarial examples. (Goldblum et al., 2019) showed that meta-learning algorithms can be supplemented with adversarial examples in the query set to learn robustness. Their method called Adversarial Querying was shown to be robust to strong attacks such as PGD. Many meta-learning approaches were extended to their robust counterparts by including adversarial query examples. Recently, (Wang et al., 2021) also proposed a similar approach where MAML was used as the base meta learning algorithm. (Wang et al., 2021) also showed that including a contrastive learning objective similar to (Chen et al., 2020) can provide a way to use unlabelled data when learning the model and thus improve both standard and robust accuracy.

## 4 EXPERIMENTS

In this section, we describe our experiments and provide implementation details. We consider three benchmark datasets - **Mini-ImageNet**, **CIFAR-FS** and **CUB** for our experiments. **Mini-ImageNet** consists of 100 classes derived from ImageNet dataset (Russakovsky et al., 2015) where each category consists of 600 images. This was first proposed in (Vinyals et al., 2016) and recent works follow the split provided by (Ravi & Larochelle, 2016) consisting of 64 base, 16 validation and 20 novel classes. We use 84x84x3 images for all experiments with Mini-ImageNet. **CIFAR-FS** was proposed in (Bertinetto et al., 2016) as a benchmark for few-shot classification. It splits CIFAR-100 dataset similar to Mini-ImageNet and we use 32x32x3 images for all our experiments. **CUB** (Wah et al., 2011) is a fine-grained dataset which has been used as a benchmark for few-shot classification. It consists of 11788 images split across 200 categories. We use the split provided by (Hilliard et al., 2018) consisting of 100 base , 50 validation and 50 novel classes. We use 84x84x3 images for experiments with CUB and as per our knowledge are the first to show adversarial robustness for a fine-grained dataset under few-shot setting. We use Pytorch (Paszke et al., 2017) framework and all our experiments use NVIDIA 2080Ti and Titan RTX GPUs.

**Implementation details:** In the base training stage, as described in Section 2.1, we follow the attack parameters of (Goldblum et al., 2019) and use iterative PGD attack with 7 iterations during training with $\epsilon = 8/255$ and $\alpha = 2/255$ for all experiments unless otherwise mentioned. We mainly use the standard network architecture ResNet (He et al., 2016) and also provide results for other architectures. For weight averaging, we set the parameter $\tau = 0.999$. We use SGD optimizer with a learning rate of 0.1 and weight decay of $1e-5$ for the feature extractor parameters $\theta_b$ and $1e-4$ for the classifier parameters $\omega_b$. We train the model for 250 epochs with a batch size of 64. For novel training and learning the Linear Classifier, we follow the setting described in (Chen et al., 2019) and learn the parameters using SGD with momentum 0.9 and learning rate $\eta = 0.01$. We set the dampening as 0.9 and weight decay of $1e-3$. For our Mean Calibration, we use $m = 2$ number of base categories. For calculating Robust Accuracy, we use 20 iterations of PGD to create an adversarial example and measure accuracy w.r.t ground truth. We report both Standard Accuracy and Robust Accuracy for 5-way, 1-shot and 5-way, 5-shot settings averaged over 1000 different trials on the test set. We report the mean of the different trials as well as the 95% confidence intervals for all our experiments. Since our goal is to build robust models, we mainly focus on improving Robust Accuracy.

We compare our results with (Goldblum et al., 2019) ,which we refer to as Adversarial Querying (AQ) where adversarial examples are created for query data and (Wang et al., 2021) which we refer as **OFA** where MAML (Finn et al., 2017) is used to adversarially train the model. We refer to training a linear classifier during the novel training stage as **Linear** and the Nearest Centroid classifier as **NC**.

## 4.1 MINI-IMAGENET

We present our results on Mini-ImageNet in Table 1. We observe that **NC** method outperforms other approaches in robust accuracy under most settings. We also show results on large-scale architectures such as WideResNets (Zagoruyko & Komodakis, 2016) and DenseNets (Huang et al., 2017). We see that our NC method outperforms on Robust Accuracy in most settings and boosts standard accuracy as well. The difference becomes clear as we move to larger architectures. Our Linear classifier also serves as a strong baseline for robust few-shot settings.

Our method is straightforward to scale for large architectures since they are equivalent to training standard architectures. However, we observed that scaling meta-learning algorithms combined with PGD adversarial training is a difficult task. To provide a comparison, we trained both AQ and our model on 4 NVIDIA TITAN RTX GPUs using WideResNet-28-10 backbones. AQ method took 1.7 hour/epoch and required 60 epochs while our method took 0.36 hour/epoch for 250 epochs. The total training time for AQ was around 100 hours whereas our method took around 90 hours. This shows that our approach is more easily scalable compared to previous methods.

| Method | Backbone | 1-shot | | 5-shot | |
|---|---|---|---|---|---|
| | | Standard Acc. | Robust Acc. | Standard Acc. | Robust Acc. |
| AQ | | $41.48 \pm 0.56$ | $20.52 \pm 0.45$ | $59.32 \pm 0.53$ | $32.18 \pm 0.50$ |
| Ours (Linear) | ResNet18 | $42.63 \pm 0.56$ | $19.56 \pm 0.45$ | $\mathbf{61.35 \pm 0.51}$ | $30.63 \pm 0.52$ |
| Ours (NC) | | $\mathbf{44.98 \pm 0.59}$ | $\mathbf{21.38 \pm 0.46}$ | $61.30 \pm 0.55$ | $\mathbf{33.41 \pm 0.51}$ |
| AQ | | $38.99 \pm 0.55$ | $22.09 \pm 0.45$ | $57.11 \pm 0.51$ | $33.62 \pm 0.50$ |
| Ours (Linear) | WRN-50-2 | $43.14 \pm 0.54$ | $19.94 \pm 0.43$ | $62.93 \pm 0.50$ | $30.52 \pm 0.52$ |
| Ours (NC) | | $\mathbf{46.71 \pm 0.62}$ | $\mathbf{23.04 \pm 0.50}$ | $\mathbf{63.60 \pm 0.55}$ | $\mathbf{36.06 \pm 0.54}$ |
| AQ | | $44.17 \pm 0.60$ | $\mathbf{23.81 \pm 0.48}$ | $62.41 \pm 0.54$ | $33.62 \pm 0.50$ |
| Ours (Linear) | WRN-28-10 | $52.36 \pm 0.62$ | $22.23 \pm 0.52$ | $\mathbf{72.11 \pm 0.51}$ | $32.29 \pm 0.59$ |
| Ours (NC) | | $\mathbf{53.22 \pm 0.66}$ | $22.91 \pm 0.51$ | $70.13 \pm 0.52$ | $\mathbf{35.40 \pm 0.58}$ |
| AQ | | $38.32 \pm 0.55$ | $10.19 \pm 0.32$ | $56.65 \pm 0.51$ | $17.77 \pm 0.41$ |
| Ours (Linear) | DenseNet121 | $39.77 \pm 0.56$ | $18.16 \pm 0.42$ | $57.45 \pm 0.54$ | $27.89 \pm 0.52$ |
| Ours (NC) | | $\mathbf{42.05 \pm 0.60}$ | $\mathbf{20.21 \pm 0.45}$ | $\mathbf{58.59 \pm 0.55}$ | $\mathbf{32.24 \pm 0.56}$ |
| AQ | | $37.35 \pm 0.52$ | $9.80 \pm 0.31$ | $55.97 \pm 0.53$ | $16.69 \pm 0.38$ |
| Ours (Linear) | DenseNet161 | $40.75 \pm 0.55$ | $17.44 \pm 0.41$ | $59.84 \pm 0.53$ | $27.11 \pm 0.50$ |
| Ours (NC) | | $\mathbf{43.48 \pm 0.60}$ | $\mathbf{20.63 \pm 0.45}$ | $\mathbf{60.92 \pm 0.54}$ | $\mathbf{33.87 \pm 0.53}$ |

Table 1: **Results on Mini-ImageNet dataset**. Our NC method outperforms other approaches which becomes clear as we move to larger architectures. We can also see that our linear classifier serves as a strong baseline and can be used to learn robust few-shot classifier.

**Comparison with OFA:** We compare with another recent work OFA (Wang et al., 2021) where MAML was combined with adversarial training to improve robustness. We present them separately compared to previous results as the attack parameters and testing configuration followed are different. For a fair comparison, we use the same setting and we refer the readers to (Wang et al., 2021). **Base training** column indicates the type of adversary used during base dataset training where AT indicates PGD adversarial training, TRADES is the algorithm presented in (Zhang et al., 2019a) and CL corresponds to using the Contrastive Learning objective (Chen et al., 2020). Both TRADES and CL use additional unlabelled data in a semi-supervised manner. We observe from Table 2 that our method has clear gains in terms of robust accuracy and surpasses standard accuracy in some cases as well.

## 4.2 CIFAR-FS

Here we present results on CIFAR-FS dataset. The organization is similar to Mini-ImageNet dataset where we presented two tables for fair comparison with previous approaches. To compare with

| Method | Base training | Conv4 | | ResNet18 | |
|---|---|---|---|---|---|
| | | Standard Acc. | Robust Acc. | Standard Acc. | Robust Acc. |
| AQ | AT | 29.6 | 24.9 | 30.04 | 20.05 |
| OFA | AT | **40.82** | 23.04 | 38.94 | 19.94 |
| OFA | TRADES | 37.1 | 25.51 | 41.94 | 20.19 |
| OFA | CL | 38.60 | 26.81 | 43.98 | 21.47 |
| Ours (Linear) | AT | $38.39 \pm 0.37$ | $28.76 \pm 0.33$ | $44.93 \pm 0.37$ | $29.30 \pm 0.33$ |
| Ours (NC) | AT | $39.23 \pm 0.38$ | **$30.77 \pm 0.35$** | **$49.15 \pm 0.41$** | **$35.59 \pm 0.38$** |

Table 2: Comparison with (Wang et al., 2021) on Mini-ImageNet dataset. Note that both TRADES and CL use additional unlabelled data in a semi-supervised manner. Our method outperforms previous approaches using a standard adversarial training procedure.

| Method | Backbone | 1-shot | | 5-shot | |
|---|---|---|---|---|---|
| | | Standard Acc. | Robust Acc. | Standard Acc. | Robust Acc. |
| AQ | | $45.41 \pm 0.68$ | $21.76 \pm 0.59$ | **$64.98 \pm 0.58$** | $34.24 \pm 0.65$ |
| Ours (Linear) | | $44.76 \pm 0.63$ | $21.01 \pm 0.58$ | $62.23 \pm 0.63$ | $31.60 \pm 0.66$ |
| Ours (NC) | ResNet18 | **$48.89 \pm 0.71$** | **$27.16 \pm 0.66$** | $64.36 \pm 0.61$ | **$39.13 \pm 0.71$** |

Table 3: **Results on CIFAR-FS dataset**. We can see our NC method outperforms compared to previous approaches. This experiment uses the same attack parameters as (Goldblum et al., 2019).

Adversarial Querying (AQ) (Goldblum et al., 2019), we use a ResNet18 backbone and use attack parameters $\epsilon = 8/255$ and $\alpha = 2/255$ with 20-iteration PGD for testing. Our results for this setting is presented in Table 10. Our Nearest Centroid classifier outperforms previous approaches on Robust Accuracy, which is the main focus of our work. We also observe that we match or outperform on Standard Accuracy under both 1-shot and 5-shot settings.

For fair comparison with (Wang et al., 2021), we train a model with Conv4 backbone and use their attack parameters. The results for this configuration is presented in Table 4. As explained earlier, **Base training** column indicates the type of adversary used during base dataset training where AT indicates PGD adversarial training, TRADES uses the algorithm presented in (Zhang et al., 2019a) and CL corresponds to using the Contrastive Learning objective (Chen et al., 2020). We see a similar trend as before where our NC outperforms previous approaches. We also conduct an experiment using TRADES during Base Training, allowing us to compare methods that use similar adversary. This experiment shows that our method can generalize to other adversarial training methods and we believe that as more advanced methods are developed in the community, they can be incorporated in a straightforward manner to improve robustness in few-shot settings.

| Method | Base training | 1-shot | | 5-shot | |
|---|---|---|---|---|---|
| | | Standard Acc. | Robust Acc. | Standard Acc. | Robust Acc. |
| AQ | AT | 31.25 | 26.34 | 52.32 | 33.96 |
| OFA | AT | 39.76 | 26.15 | 57.18 | 32.62 |
| OFA | TRADES | 40.59 | 28.06 | 57.62 | 34.76 |
| OFA | CL | 41.25 | **29.33** | **57.95** | 35.3 |
| Ours (Linear) | AT | $41.12 \pm 0.40$ | $25.65 \pm 0.37$ | $56.20 \pm 0.39$ | $34.73 \pm 0.41$ |
| Ours (NC) | AT | $41.81 \pm 0.41$ | $28.22 \pm 0.40$ | $53.52 \pm 0.40$ | $39.09 \pm 0.42$ |
| Ours (NC) | TRADES | **$43.56 \pm 0.43$** | $28.12 \pm 0.41$ | $56.99 \pm 0.40$ | **$39.48 \pm 0.43$** |

Table 4: Comparison with (Wang et al., 2021) for Conv4 backbone on CIFAR-FS dataset. Comparing methods that use same base training procedure (**AT** or **TRADES** ) , we can see that our NC method outperforms on Robust Accuracy under both 1-shot and 5-shot settings. This experiment shows that our method can generalize to other adversarial training methods as well.

## 4.3 CUB

For results on CUB dataset, we use the same attack parameters described for Mini-ImageNet i.e, $\epsilon = 8/255$, $\alpha = 2/255$, 7 iterations of PGD during training and 20 during testing. We use ResNet18

backbone and implement AQ as per the guidelines given in (Goldblum et al., 2019) and our best implementation with hyperparameters is presented in Table 5.

Since CUB is a fine-grained classification dataset, the base and novel categories share greater similarity compared to previous datasets. Hence it serves as an opportunity to understand how the robustness transfers from base to novel dataset, i.e whether the similarity in classes acts as a boon or bane under fine-grained dataset settings. As can be seen from our experiments, the linear classifier baseline performs reasonably well compared to previous approach indicating that a robust base classifier transfers to a robust novel classifier. The NC method also benefits under such settings and outperforms all other methods on both Standard and Robust Accuracy.

| Method | Backbone | 1-shot | | 5-shot | |
|---|---|---|---|---|---|
| | | Standard Acc. | Robust Acc. | Standard Acc. | Robust Acc. |
| AQ | | $54.27 \pm 0.79$ | $28.23 \pm 0.66$ | $68.42 \pm 0.62$ | $37.10 \pm 0.66$ |
| Ours (Linear) | | $51.93 \pm 0.71$ | $27.24 \pm 0.64$ | $69.83 \pm 0.61$ | $37.06 \pm 0.68$ |
| Ours (NC) | ResNet18 | $\mathbf{56.42 \pm 0.78}$ | $\mathbf{32.18 \pm 0.70}$ | $\mathbf{71.51 \pm 0.60}$ | $\mathbf{44.33 \pm 0.69}$ |

Table 5: **Results on CUB dataset**. We show that robustness transfers from base to novel datasets under fine-grained classification setting as well. Our Linear classifier serves as a strong baseline and our NC method outperforms on both metrics.

## 5 DISCUSSION

In this section, we provide additional intuition into our method and also showcase some benefits of our simple framework. We also refer the readers to appendix for more experiments and analysis.

**Understanding the effect of each component :** Our proposed approach consists of multiple components and we would like to study the effect of each of those. We perform an experiment on Mini-ImageNet dataset on ResNet18 backbone similar to Table 1 and the results are presented in Table 6. We observe that the simple baseline of robust base training and training a linear classifier during novel training works reasonably well which can be a considered a strong baseline (Exp Id 1,4). We also observe that DC algorithm proposed in (Yang et al., 2021) improves standard accuracy but introduces a drop in robustness (Exp Id 2,5). In most configurations we observe that our method matches or even outperforms the DC method and more importantly improves robustness (Exp Id 5,6). This shows that the robustness of the base dataset is transferred to the novel setting and the mean calibration along with Nearest Centroid classifier can improve robustness in few-shot setting. The impact of weight averaging (WA) method can be observed when considering Exp Ids 3 and 6. The temporal ensembling nature of the method helps in finding flatter minima, thereby boosting performance. Previous works (Gowal et al., 2020) have shown that this can be used for standard robust classifiers. Here, we observe this holds true under a transfer-learning type setting. We find that the combination of these methods can lead to improved performance.

| Exp. Id | Base Training WA | Novel Training DC | NC | Linear | 1-shot Standard Acc. | Robust Acc. | 5-shot Standard Acc. | Robust Acc. |
|---|---|---|---|---|---|---|---|---|
| 1 | ✗ | ✗ | ✗ | ✓ | $41.40 \pm 0.56$ | $18.25 \pm 0.45$ | $\mathbf{59.30 \pm 0.54}$ | $27.96 \pm 0.50$ |
| 2 | ✗ | ✓ | ✗ | ✗ | $\mathbf{43.72 \pm 0.57}$ | $14.18 \pm 0.38$ | $58.04 \pm 0.52$ | $13.97 \pm 0.39$ |
| 3 | ✗ | ✗ | ✓ | ✗ | $42.56 \pm 0.60$ | $\mathbf{19.57 \pm 0.45}$ | $58.22 \pm 0.53$ | $\mathbf{30.42 \pm 0.50}$ |
| 4 | ✓ | ✗ | ✗ | ✓ | $42.63 \pm 0.56$ | $19.56 \pm 0.45$ | $\mathbf{61.35 \pm 0.51}$ | $30.63 \pm 0.52$ |
| 5 | ✓ | ✓ | ✗ | ✗ | $44.73 \pm 0.59$ | $15.29 \pm 0.41$ | $59.78 \pm 0.53$ | $19.49 \pm 0.49$ |
| 6 | ✓ | ✗ | ✓ | ✗ | $\mathbf{44.98 \pm 0.59}$ | $\mathbf{21.38 \pm 0.46}$ | $61.30 \pm 0.55$ | $\mathbf{33.41 \pm 0.51}$ |

Table 6: Illustration of different configurations of Base and Novel training. Here we show results on ResNet18 backbone on Mini-ImageNet dataset. WA represents Weight Averaging, DC represents Distribution Calibration and NC corresponds to to Nearest Centroid classifier.

**Extension to verifiably robust models:** An advantage of our simple framework is that we can incorporate methods from the adversarial examples literature for few-shot learning. Specifically,

we consider verifiably robust procedures where the goal is to provide a guarantee on adversarial robustness of the model. Standard adversarial training methods do not lead to provably robust models leading to low verified accuracy, we observe a similar trend in our experiments as well. Many methods have been proposed in this area (Mirman et al., 2018; Zhang et al., 2019b; Cohen et al., 2019) and we consider one of them - Interval Bound Propagation (IBP) (Gowal et al., 2018). Here we show that replacing PGD-training with IBP during the robust base training stage described in section 2.1 can lead to verifiable robustness for few-shot classifiers. We show results for 1-shot setting in Table 7 where we use a ResNet18 backbone on CIFAR-FS dataset. We use the training procedure as described in (Xu et al., 2020) with $\epsilon = 8/255$ for 1000 epochs. Here Robust Accuracy refers to 20-iteration PGD testing and Verified Accuracy is calculated similar to (Gowal et al., 2018). We believe this experiment can encourage researchers to incorporate more advanced verification methods in the future and also develop algorithms specific to few-shot settings.

| Method | 1-shot | | |
|---|---|---|---|
| | Standard Acc. | Robust Acc. | Verified Acc. |
| IBP + Linear | $37.01 \pm 0.65$ | $26.77 \pm 0.59$ | $21.79 \pm 0.55$ |
| IBP + NC | $37.72 \pm 0.65$ | $28.12 \pm 0.62$ | $23.25 \pm 0.61$ |

Table 7: Extension to verifiably robust classifiers. We show that it is possible to train verifiably robust models for few-shot settings. This is an added advantage of our framework due to the similarity to standard classifier training. Results are shown using ResNet18 backbone on CIFAR-FS dataset.

**Using Adversarial examples from few-shot categories:** We additionally consider an experiment where adversarial examples of the few-shot data are used when learning the linear classifier. Although this makes the learning more complex due to the adversarial training procedure, we perform this baseline experiment and learn the linear layer. As seen in Table 8 we observe a drop in performance which suggests that the network might be biased towards adversary of the few-shot data. We use a 7-step PGD and 20-step PGD when learning the classifier, similar to adversarial training. This suggests that creating adversarial examples increases computation but does not lead to performance gains. Our NC method does not include adversarial examples and outperforms previous methods without increasing computational burden. This experiment uses ResNet18 backbone on Mini-ImageNet dataset.

| Method | 1-shot | |
|---|---|---|
| | Standard Acc. | Robust Acc. |
| Linear No Adv | $42.63 \pm 0.56$ | $19.56 \pm 0.45$ |
| Linear 7-PGD | $42.00 \pm 0.56$ | $18.83 \pm 0.42$ |
| Linear 20-PGD | $42.03 \pm 0.58$ | $19.01 \pm 0.42$ |
| Ours (NC) | $\mathbf{44.98 \pm 0.59}$ | $\mathbf{21.38 \pm 0.46}$ |

Table 8: Experiment where adversarial examples from few-shot data are included to perform adversarial training on the linear classifier. **No Adv** refers to clean examples used for learning the linear layer. **7-PGD** refers to 7-step PGD and **20-PGD** refers to 20-step PGD used in training. Robust Accuracy is calculated using 20-step PGD. Our NC method outperforms on both metrics without increasing computation due to adversarial training.

## 6 CONCLUSION AND FUTURE WORK

We present a simple and scalable approach for improving robustness in few-shot image classifiers. Our method outperforms previous approaches when compared with standard few-shot learning benchmarks on both standard and robust accuracy. Note that our method is similar to traditional adversarial machine learning approaches rather than meta-learning methods, hence it becomes easier to introduce concepts such as certified robustness which is unexplored for few-shot classifiers. We show preliminary results in this direction and plan to explore this direction in future works. We believe that the simplicity of our approach would be beneficial to the community and upon which researchers can develop robust few-shot classifiers.

## 7 ETHICS STATEMENT AND REPRODUCIBILITY

We believe most AI algorithms including ours can be exploited by adversaries for unethical applications. On the positive side, we are developing algorithms that make deep networks robust to adversarial attacks, eliminating some of such undesired applications. Moreover, while publishing such defense algorithms may enable a larger community to access and benefit from them, leading to democratized AI, it can also enable adversaries to develop better attacks.

To ensure that our results are reproducible, we include all implementation details of our work and also provide code. We believe that readers will have sufficient knowledge of the hyperparameters associated with our method to reproduce the results.

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

## A APPENDIX

**Variation of Robust Accuracy with number of attack iterations:** We vary the number of attack iterations of PGD and observe a fairly stable performance for both 1-shot and 5-shot settings, as seen in Table 9. This shows that defense performs well and is not sensitive to the number of attack iterations. We perform this experiment on Mini-Imagenet dataset with ResNet-12 backbone.

| PGD Iterations | 1-shot Robust Acc. | 5-shot Robust Acc. |
|---|---|---|
| 20 | $25.32 \pm 0.52$ | $38.83 \pm 0.57$ |
| 40 | $25.19 \pm 0.53$ | $38.46 \pm 0.54$ |
| 100 | $25.64 \pm 0.53$ | $38.22 \pm 0.56$ |
| 200 | $25.09 \pm 0.54$ | $38.62 \pm 0.57$ |

Table 9: Variation of attack iterations

**Variation of Robust Accuracy with perturbation budget $\epsilon$:** To check for the absence of gradient masking, we increase $\epsilon$ from 8/255 to 128/255 in Figure 1. As expected, we observe that both 1-shot and 5-shot accuracy drop to zero with increased $\epsilon$. Note that we plot only the the mean accuracy over 1000 different tasks.

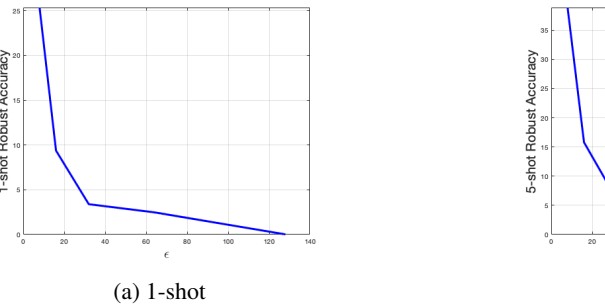

(a) 1-shot      (b) 5-shot

Figure 1: Variation of Robust Accuracy with different perturbation budget $\epsilon$

**Variation of Robust Accuracy with number of base neighbors $m$:** We plot the variation of Robust Accuracy with number of base neighbors $m$ in Figure 2. We find best results with $m = 2$ which we use for all our experiments. Note that we plot only the mean accuracy over 1000 different tasks.

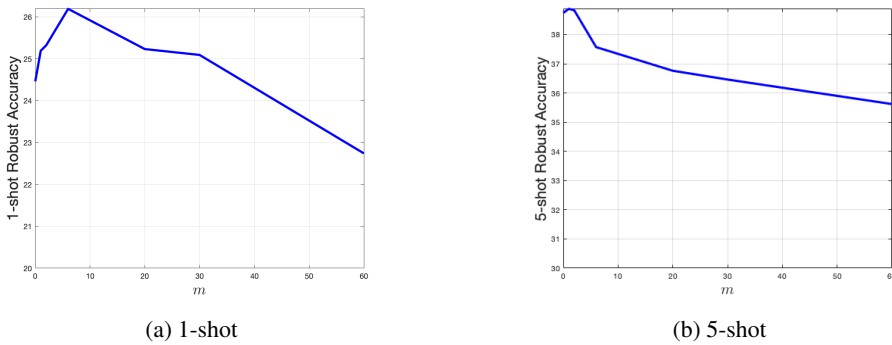

(a) 1-shot      (b) 5-shot

Figure 2: Variation of Robust accuracy with number of base centers $m$ for 1-shot and 5-shot settings

**Number of base categories chosen:** We plot the variation of standard accuracy with $m$ in Figure 3 for 1-shot and 5-shot settings. It is to be noted that we plot only the mean accuracy averaged over

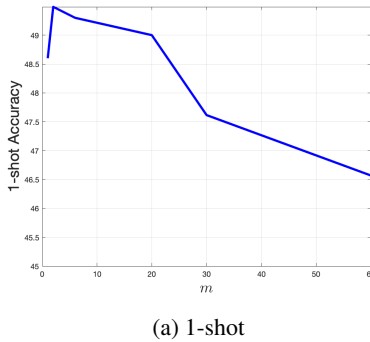

(a) 1-shot

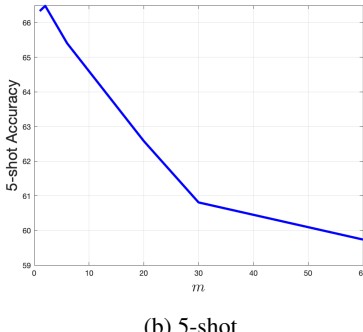

(b) 5-shot

Figure 3: Variation of Standard accuracy with number of base centers $m$ for 1-shot and 5-shot settings

1000 test episodes. We find best results with $m = 2$ which we use for all our experiments.

**Results on TieredImageNet:** We also consider TieredImageNet(Ren et al., 2018), which is larger subset of ILSVRC12, sampled from hierarchically. We use the standard split of 351 base classes and 160 novel classes. We see once again that the linear classifier acts as a strong baseline and our NC method outperforms previous method. This is an interesting experiment compared to the one using Mini-ImageNet which is also sampled from ILCVRC12 and shows that our method works even when the similarity between base and novel classes is varied.

| Method | Backbone | 1-shot | | 5-shot | |
|---|---|---|---|---|---|
| | | Standard Acc. | Robust Acc. | Standard Acc. | Robust Acc. |
| AQ | | $49.77 \pm 0.70$ | $29.78 \pm 0.65$ | $66.72 \pm 0.56$ | $43.73 \pm 0.63$ |
| Ours (Linear) | ResNet18 | $50.47 \pm 0.69$ | $27.90 \pm 0.60$ | $68.48 \pm 0.60$ | $40.30 \pm 0.65$ |
| Ours (NC) | | $51.38 \pm 0.71$ | $30.27 \pm 0.62$ | $68.50 \pm 0.59$ | $44.64 \pm 0.66$ |

Table 10: Results on TieredImageNet dataset.

