# OpenReview forum: "A Simple Approach to Adversarial Robustness in Few-shot Image Classification "
_ICLR.cc/2022/Conference — ICLR 2022 Submitted_

### Official Review · Reviewer_nGCx · 2021-10-17

**Correctness:** 3
**Technical Novelty And Significance:** 2
**Empirical Novelty And Significance:** 2
**Recommendation:** 5
**Confidence:** 2

**Main Review:**

1) What is the abundantly-labeled base dataset $X_b$ used in this work? How do you guarantee that there is no overlap information between $X_b$ and $X_n$.

2) It is important to under standard each stage i.e., base training and novel training performance. However, I have not seen ablation uses base training only and novel training only.


**Summary Of The Paper:**

This paper 1) proposes a simple transfer learning approach to enable train adversarially robust few-shot classifiers for few-shot image classification, and 2) present a method for novel classification task based on calibrating the centroid of the few-shot category towards the base classes. Results show good performance has been achieved on three benchmarks i.e., Mini-ImageNet, CIFAR-FS, and CUB datasets.

My main concern of this work is that the improved performance can be mainly resulted from the pretrained stage (base training) using the base dataset $X_b$. Normally, pretraining can improve the performance on small tasks.

**Summary Of The Review:**

My main concern of this paper is the comparison maybe not fair i.e., this work uses a pretrained stage. It is also difficult for us to characterize if any overlap information between $X_b$ and $X_n$. Overall, I think it should compare all methods in the same condition. The ablation of base training and novel training is necessary to characterize the importance of each stage for robust few-shot learning.

---

> ### Author Response · Authors · 2021-11-18
> **Author Response**
>
> We thank the reviewer for the detailed feedback and questions. We answer them below and please let us know if you have any further questions regarding the dataset settings or in our experiments.
>
> $\textbf{``What is the abundantly-labeled base dataset?''}$- We use the same dataset $X_b$ considered in standard few-shot learning literature, hence providing for a fair comparison.  For Mini-ImageNet, we use the split provided by [3] , for CIFAR-FS we use the split provided by [4] and for CUB we use the split provided by [5]. We can guarantee that there is no overlap of class information between $X_b$ and $X_n$ since we consider a disjoint set of classes as base and novel before sampling from them.
>
> $\textbf{``It is important to understand each stage''}$ - We agree with the reviewer that the effect of both stages should be studied. We present one such ablation in Table 6 where we vary different components for Base and Novel Training. Note that since Base and Novel datasets contain different classes, we need to adapt the Base training model using one of the methods (Linear classifier or Nearest Centroid classifier) for evaluating the adaptation to classes from the Novel dataset.
>
>
> $\textbf{References:}$
>
> [1] - Adversarially Robust Few-Shot Learning: A Meta-Learning Approach, Goldblum et al., NeurIPS 2020
>
> [2] - On Fast Adversarial Robustness Adaptation in Model Agnostic Meta-Learning, Wang et al., ICLR 2021
>
> [3] - Optimization as a model for few-shot learning. Ravi et al., ICLR 2017
>
> [4] - Learning feed-forward one-shot learners, Bertinetto et al., NeurIPS 2016
>
> [5] - Few-shot learning with metric-agnostic conditional embeddings. Hilliard et al., arXiv:1802.04376

---

### Official Review · Reviewer_QZee · 2021-10-19

**Correctness:** 3
**Technical Novelty And Significance:** 2
**Empirical Novelty And Significance:** 1
**Recommendation:** 5
**Confidence:** 4

**Main Review:**

(+) The paper is well written and organized and easy to follow.
(+) The proposed approach is simple and technically sound.
(+) Multiple benchmarks have been employed to validate the performance.

(-) The proposed method is a straightforward combination of well-known knowledge and techniques. Although it is technically sound, it is mainly empirically designed without significant contribution to the field.
(-) The experiments are limited as:
1. all the three benchmarks are toy-level datasets;
2. only two baselines are compared that can hardly represent the current state of the arts;
3. A single metric of accuracy is used for all the experiments which failed to justify the proposed method from a different perspective.
4. The performance increase is either negative or very subtle in general.


**Summary Of The Paper:**

This paper proposes a method to train adversarially robust few-shot classifiers. The main idea is to apply adversarial training (PGD) to pre-train the feature extractor and apply Nearest Centroid to update the classifier with novel data for few-shot learning.

**Summary Of The Review:**

To sum up, this is a mediocre submission and I would rank it below the acceptance bar of ICLR.

---

> ### Author Response · Authors · 2021-11-18
> **Author Response**
>
> We thank the reviewer for the detailed feedback and we address the concerns in our work below. Please let us know if we could answer any further questions regarding the method or experiments.
>
>
> $\textbf{``All three benchmarks are toy-level datasets'' }$ - We respectfully disagree with the reviewer as all three datasets have been accepted as large-scale benchmarks in the few-shot learning community. Mini-ImageNet [1] in particular is derived from the ImageNet dataset and is considerably challenging task. This is why we do not consider simpler datasets such as Omniglot[2] where standard accuracies have saturated ($\sim$ 96 \%). We believe that all three datasets present their own unique challenges under few-shot setting and hence are valid benchmarks to consider.
> Moreover, we ran an experiment on tieredImageNet [3] which is a larger subset of ImageNet and has categories split in a different manner. We present our results compared with AQ [4] in the table below.
>
> | Method  | Backbone   | 1-shot Standard Acc |1-shot Robust Acc |5-shot Standard Acc |5-shot Robust Acc |
> | ----------- | ----------- |----------- |----------- |----------- |----------- |
> |AQ | ResNet18  | 49.77 $\pm$0.70 |  29.78 $\pm$ 0.65 | 66.72 $\pm$ 0.56 | 43.73 $\pm$ 0.63  |
> |Ours (NC) |   | 51.38 $\pm$ 0.71 | 30.27 $\pm$ 0.62    | 68.50 $\pm$  0.59 | 44.64 $\pm$ 0.66 |
>
>
> $\textbf{`` Only two baselines are compared '' }$-  Adversarial Robustness for few-shot setting is a relatively new problem and there have been few works that have tried to study this. We compare with the recently published works AQ[4] (published in NeurIPS 2020, December) and OFA [5] (published in ICLR 2021, May). Hence our comparison with state-of-the-art methods is fair.
>
> $\textbf{`` Performance increase is either negative or very subtle in general''}$ - We would like to point out that the main novelty of our method is that we are able to achieve better performance as previous approaches in a simplistic manner. This allows for multiple benefits such as improved training time, scalability and also the fact that these can be extended to verifiably robust training. We view the improved performance as one of the contributions rather than the main contribution. Hence we respectfully disagree with the reviewer that ours is a mediocre submission. We believe that our method can benefit researchers to improve upon and build more empirical and verifiably robust models for few-shot setting.
>
>
> $\textbf{`` A single metric of accuracy is used for all experiments '' }$ -
> We consider both 1-shot and 5-shot accuracy for both standard and robust  settings. We believe this is common practice for evaluating robust classification models. Please feel free to suggest other suitable metrics and we will be happy to discuss it.
>
> $\textbf{References:}$
>
> [1] - Matching Networks for One Shot Learning, Vinyals et al. NeurIPS 2016
>
> [2] - The Omniglot challenge: a 3-year progress report, Lake et al.
>
> [3] - Meta-Learning for Semi-Supervised Few-Shot Classification, Ren et al., ICLR 2018
>
> [4] - Adversarially Robust Few-Shot Learning: A Meta-Learning Approach, Goldblum et al., NeurIPS 2020

---

### Official Review · Reviewer_iADS · 2021-11-02

**Correctness:** 4
**Technical Novelty And Significance:** 2
**Empirical Novelty And Significance:** 3
**Recommendation:** 6
**Confidence:** 4

**Main Review:**

Strength:
1. The paper is simple to follow and clear for the reader to understand.
2. The experiment is valid and cover various ablation study.

Weakness
1. This paper demonstrates a simple approach for learning a robust few shot classifier. Despite its simplicity, the contribution of this paper remains weak and vague. To my best knowledge, this paper has minor technical novelty, because the few shot learning method used in this work have already existed in the few shot literature. The adversarial training used in this work also follow the standard adversarial training manner. As a result, the technical contribution of this work remains unclear.
2. While this paper demonstrates a simple baseline for learning an adversarial robust few shot classifier, there is no theoretical analysis of why such simple manner is working. From my best understanding, the only support of this paper is the experiment result, which is only evaluated on 2 different few shot approach. It is unclear whether the proposed training scheme can generalize to other few shot learning method.   Moreover, the main experiments are conducted using PGD and IBP attack/adversarial training, indicating the generalization of this framework to other attack method is questionable.


**Summary Of The Paper:**

This paper aims to address the problem of adversarial attack for low shot image classification. This work is motivated by the challenging scenario where there is a need of significant amount of data to train an adversarial robust classifier and there is not much data under few-shot setting. This work proposed and demonstrated a simple approach for robust few shot classifier. A model is first adversarially trained on the base classes and produce a robust base model. The feature extractor of the robust base model is then frozen. With the frozen feature extractor, 2  different manners are used for training a novel / few shot classifier, including (1) training a linear classifier and (2) computing category centroid for each novel class and perform nearest neighbor classification using the centroid. The experiment demonstrates that this simple baseline can outperform prior baselines on 3 datasets. The ablation study includes 1 vs 5 shot and different adversarial training methods. However, the core contribution of this paper is weakened due to the lack of theoretical analysis.

**Summary Of The Review:**

While this paper demonstrates good results with the proposed simple framework, there is no theoretical analysis of why such simple method is working. Moreover, it is also unclear to the reader whether the proposed framework generalize to other attack methods and other few shot approaches.

---

> ### Author Response · Authors · 2021-11-18
> **Author Response**
>
> We thank the reviewer for a detailed feedback. We provide clarifications to the questions below, please let us know if you have any further questions regarding our method or experiments.
>
>
> $\textbf{``Unclear if proposed framework generalizes to other attack methods''}$- That is a great question, and although we show results using PGD and IBP in the paper, we add another experiment in the table below where we use TRADES [1] algorithm for base adversarial training on CIFAR-FS dataset and compare it with OFA[2]. Note that we also incorporate the additional unlabeled data used in OFA[2].
> As seen below, our method outperforms OFA by more than 4 points on 5-shot setting. These results are similar to Table 4 and will be added to it in the revised draft. Since we show results using 3 different adversarial base training procedures, we believe that our method will also generalize to more advanced adversarial training procedures developed by the community in the future.
>
> | Method  | Base Training    | 1-shot Standard Acc |1-shot Robust Acc |5-shot Standard Acc |5-shot Robust Acc |
> | ----------- | ----------- |----------- |----------- |----------- |----------- |
> |OFA | TRADES   | 40.59 |  28.06 | 57.62 | 34.76  |
> |Ours (NC) | TRADES  |43.56 $\pm$ 0.43 | 28.12 $\pm$ 0.41    | 56.99 $\pm$  0.40 | 39.48 $\pm$ 0.43 |
>
>
>
>
>
> $\textbf{``Contribution is weak ''}$ - Our main contribution is that a simple method involving standard adversarial training combined with Mean Calibration and Nearest Centroid classifier results in better or on-par performance compared to SOTA methods that involve sophisticated methods, e.g., meta-learning. This results in multiple benefits-
>
>
> &nbsp;&nbsp;&nbsp;&nbsp;&nbsp;&nbsp;1. Our simple method is more scalable to larger architectures compared to meta-learning based methods since those methods involve calculating the second derivative that is expensive in terms of computation and memory. This challenge was discussed in OFA [2] where different approximations were considered to make the training easier. The simplicity of our method allows for easier training without any approximations.
>
> &nbsp;&nbsp;&nbsp;&nbsp;&nbsp;&nbsp;2. It also allows us to explore verification based methods which provide verifiably robust models (instead of empirically robust) as demonstrated by our usage of IBP for base training. To the best of our knowledge, ours is the first work that considers verification for few-shot settings and the simple nature of our approach is the reason behind this. It becomes non-trivial to extend verification based methods for meta-learning approaches and our framework shows that it may not be necessary.
>
>
>
> $\textbf{``Theoretical Analysis''}$ - Although we do not provide a theoretical analysis of our work, we do provide some intuition behind it. This is briefly described in the 2nd paragraph of Page 2. Recent works [7,8] have shown that models that see large number of examples from all categories get a better understanding of class semantics, which makes it easier to learn new classes quickly. Previous works have shown this for standard accuracy, we show that considering such a setting can be  beneficial for adversarial robustness as well. Note that the simplicity of our approach and similarity to standard training procedures can also benefit researchers to develop theoretical analysis for few-shot settings, which is beyond the scope of current work.
>
>
> $\textbf{``Only evaluated on 2 different few-shot approach''}$ - We would like to point out that AQ [3] considered different few-shot learning methods such as Prototypical networks [4], Meta OptNet[5] etc. and found that the R2D2[6] method outperforms other few-shot learning approaches. We compare with the best results of AQ and show improved performance.
>
>
>
> $\textbf{References:}$
>
> [1] - Theoretically Principled Trade-off between Robustness and Accuracy, Zhang et al. ICML 2019
>
> [2] - On Fast Adversarial Robustness Adaptation in Model-Agnostic Meta-Learning, Wang et al., ICLR 2021
>
> [3] - Adversarially Robust Few-Shot Learning: A Meta-Learning Approach, Goldblum et al., NeurIPS 2020
>
> [4] - Prototypical Networks for Few-shot Learning, Snell et al., NeurIPS 2017
>
> [5] - Meta-Learning with Differentiable Convex Optimization, Lee et al., CVPR 2019
>
> [6] - Meta-learning with differentiable closed-form solvers , Bertinetto et al., ICLR 2019
>
> [7]-  A Closer Look at Few-shot Classification, Chen et al., ICLR 2019
>
> [8] - A Baseline for Few-Shot Image Classification, Dhillon et al., ICLR 2019

---

### Author Response · Authors · 2021-11-18
**General Response to all reviewers**

We thank all the reviewers for the detailed feedback and we summarize our contributions as follows.

Recently adversarial robustness for few-shot learning has received attention. Previous approaches have tried to use meta-learning based methods combined with adversarial training. This is computationally expensive due to the bi-level optimization procedure.

$\textbf{Our  method:}$ Our main method involves simplifying the approach towards achieving robustness under few-shot settings. We perform adversarial training on the base dataset similar to the standard training procedure and then propose two approaches when adapting to the novel task. We show that a simple linear classifier serves as a strong baseline. We also present Mean Calibration, which involves moving the centroid of the few-shot categories towards the base categories and then we use a Nearest Centroid classifier which improves robustness. We show improved performance compared to state-of-the-art meta learning based methods across multiple standard benchmarks such as MiniImageNet, CIFAR-FS and CUB datasets.

${\bf Significance:}$ We believe our method has two advantages.

&nbsp;&nbsp;&nbsp;&nbsp;&nbsp;&nbsp;(1) It becomes easier to scale to larger architectures due to the simplicity of our approach. This is demonstrated by the reduced training time taken by our method compared to previous approaches.

&nbsp;&nbsp;&nbsp;&nbsp;&nbsp;&nbsp;(2) We also show that our framework makes it easier to develop verifiably robust networks for few-shot settings. We show that by replacing the PGD adversarial training during the base stage with Interval Bound Propagation, we are able to develop verifiably robust classifiers for few-shot task. To the best of our knowledge, this is the first attempt at extending verification for few-shot settings. We believe the simplicity of our approach is a strong contribution allowing for easier understanding of adversarial robustness for few-shot algorithms and will benefit researchers to develop both empirical and verifiable approaches of adversarial training under few-shot settings.

---

### Author Response · Authors · 2021-11-23
**Updated draft**

We would like to once again thank the reviewers for their detailed feedback. We have updated the draft to include the new experiments on TieredImageNet dataset and the experiment using TRADES for base training. Please let us know in case you have any questions or comments.

---

### Author Response · Authors · 2021-11-30
**Requesting a response to our rebuttal**

We thank the reviewers once again for their detailed feedback. Please let us know in case there are any questions/comments. Since this is the last day of the discussion period, we are hoping to receive a response from reviewers soon.

---

### Decision · Program_Chairs · 2022-01-20

**Decision:**

Reject

**Comment:**

This paper finally received divergent and borderline reviews with one positive (6) and two negative (5) rates. Based on the reviews, authors’ responses and updated manuscript, we would like to decide to reject this work at this time even though this submission has a lot of potentials such as simplicity and efficiency.
Positively, all the reviews agree that the proposed approach is simple but effective to improve the robustness of few-shot classifiers. However, there is some room for improvement to be a stronger submission: (i) the technical novelty may need to be better presented, and (ii) the improved performance may need to be better justified (e.g., the effect of the pretrained stage).